# Income distribution in Thailand is scale-invariant

**Thitithep Sitthiyot**[ID]*

Department of Banking and Finance, Faculty of Commerce and Accountancy, Chulalongkorn University, Bangkok, Thailand

* thitithep@cbs.chula.ac.th

## Abstract

This study examines whether income distribution in Thailand has a property of scale invariance or self-similarity across years. By using the data on income shares by quintile and by decile of Thailand from 1988 to 2021, the results from 306-pairwise Kolmogorov-Smirnov tests indicate that income distribution in Thailand is statistically scale-invariant or self-similar across years with *p-values* ranging between 0.988 and 1.000. Based on these empirical findings, this study would like to propose that, in order to change income distribution in Thailand whose pattern had been persisted for over three decades, the change itself cannot be gradual but has to be like a phase transition of substance in physics.

## Introduction

Thailand has made a remarkable progress in poverty reduction during the past three decades [1]. Nevertheless, income distribution in Thailand is skewed towards a small percentage of the population [2]. The recent data on income statistics reported by the National Economic and Social Development Council (NESDC) [3] indicate that, in 2021, the income share of the top quintile accounts for 50% of total income share whereas the income shares of the bottom four quintiles accounts for 50% of total income share. The income distribution is even more skewed towards a small percentage of the population when focusing only on the income shares among the top two deciles. According to the same dataset on income statistics from the NESDC [3], the income share of the $10^{th}$ decile is equal to 67% of the income shares of the top two deciles while the income share of the $9^{th}$ decile is equal to 33% of the income shares of the top two deciles. Despite the socio-economic development of Thailand has advanced the country in many aspects, creating occupational and income securities which in turn lead to poverty reduction, the NESDC [4] notes that the progress in resolving the issue with regard to income distribution has been somewhat slow as reflected by the pattern of income distribution that had not markedly changed over the past three decades. Figs 1 and 2 illustrate the patterns of income shares by quintile (Q) and by decile (D) of Thailand from 1988 to 2021 created by using the data on income statistics obtained from the NESDC [3].

While the issue of income distribution needs to be tackled urgently through various policies and measures so that it does not become an impediment to Thailand's economic and social development towards the Sustainable Development Goals (SDGs) [4], a large separate body of

**Data Availability Statement:** All relevant data are within the paper and its Supporting Information file. They are also publicly available and can be accessed from the Office of the National Economic and Social Development Council website (https://www.nesdc.go.th/main.php?filename=PageSocial).

**Funding:** The author received no specific funding for this work.

**Competing interests:** The author has declared that no competing interests exist.

research, especially in the fields of complexity economics and econophysics, have consistently shown that income distribution has a property of scale invariance or self-similarity. Generally speaking, a distribution of size is said to have a property of scale invariance or self-similarity if its essential structural and/or dynamical properties remain unchanged when considering such a distribution at different scales [5]. In other words, the distribution of size is the same whatever the scale we are looking at [6]. While scale invariance is not a common term in economics and an expression like self-similarity is scarcely used [7], the empirical evidence on scale invariance in income and wealth distributions could be dated back to the works by Pareto [8, 9] who observes that income distribution varies very little in space and time in that different people and different eras yield very similar results. Pareto [9] also notes that the shape of income distribution is remarkably stable. In addition, Mandelbrot [10] observes that over a

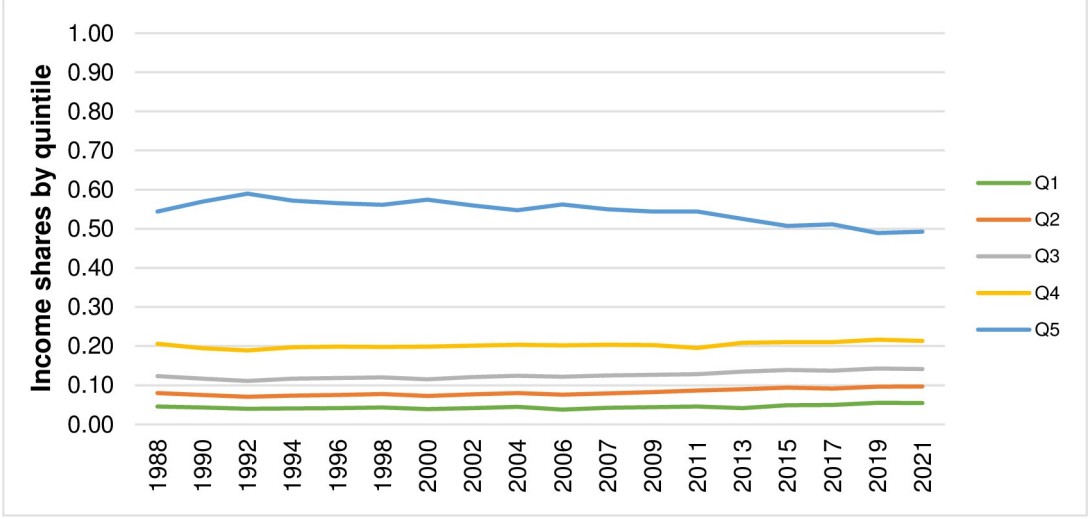

**Fig 1. Patterns of income shares by quintile of Thailand from 1988 to 2021.**

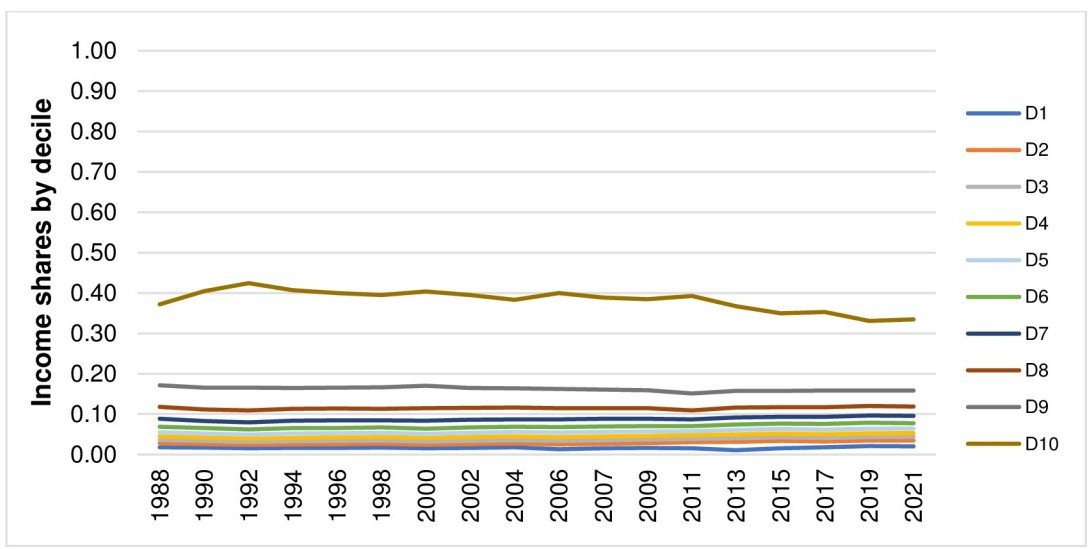

**Fig 2. Patterns of income shares by decile of Thailand from 1988 to 2021.**

certain range of values of income, its distribution is not markedly influenced either by socio-economic structure of the community under the study or by definition of income being chosen. Moreover, Newman [11] and Klass et al. [12] investigate the distribution of wealth of the richest published by *Forbes* magazine and find that the wealth distribution has a property of scale invariance. According to Anderson [13], this implies that the same dynamical rules of gains and losses apply across the entire economy independently of particular sector or wealth and sophistication of different individuals. Furthermore, Chakrabarti et al. [14] review literatures on income and wealth distributions and conclude that income and wealth distributions follow a universal pattern irrespective of differences in culture, history, social structure, indicators of prosperity, and economic policies adopted in different countries. For more than a century, numerous studies have investigated and confirmed that income and wealth distributions have no characteristic scale (see Jagielski et al. [15] and references therein). A recent study by Sitthiyot et al. [16] confirms previous empirical findings in that the distribution of average executive compensation is statistically scale-invariant or self-similar across time period, industry type, and company size. That is time period, type of industry, and company size have no effects on the distribution of average executive compensation.

The property of scale invariance is not only found in income and wealth distributions but also in many distributions of sizes that occur in nature and society. Mandelbrot [17] and Fama [18] show that the distributions of changes in stock prices are scale-invariant whereas Zajdenweber [19] finds scale invariance in the distribution of business interruption insurance claim sizes. In addition, Axtell [20] studies the distribution of firm sizes in the United States of America and finds that it has the property of scale invariance in that it does not vary across time periods and definitions of firm size being used. Moreover, Barabási [21] shows that the distributions of many networks of major scientific, technological, and societal importance such as internet, protein interactions, emails, and citations have scale invariance property. Furthermore, Broido and Clauset [22] investigate almost 1000 social, biological, technological, transportation, and information networks and find that there are differences in degree of scale invariance among these networks. According to Bak [23], West and Brown [24], and West [25], scale invariance reflects underlying generic features and physical principles that are independent of detailed dynamics or specific characteristics of particular systems.

To examine the property of scale invariance or self-similarity in income distribution, the method typically employed by previous studies [8, 10, 14, 15, 26–30] is to conduct complicated statistical analyses to find out whether income distribution follows power law, gamma, or log-normal. This is because if the data on income could be fitted by either of these three distributions, it implies that the distribution of income is scale-invariant. According to Chakrabarti et al. [14], there tends to be an agreement among academic scholars in economics, statistics, and physics that the upper tail of income distribution could be well described by power law but the lower part of income distribution could be fitted by power law, gamma, or log-normal distribution.

Given that there has yet to be an agreement on the choice of statistical distributions to be used in order to fit the data on income and test the property of scale invariance, this study therefore employs an alternative method that is based on the Kolmogorov-Smirnov test (K-S test) introduced by Sitthiyot et al. [16] which is relatively simple and does not require a priori assumption with regard to the distribution of data. To our knowledge, no study has conducted a formal test whether the distribution of income in Thailand is statistically scale-invariant across years before. From our viewpoint, the link between scale invariance and income distribution is of importance and worth investigating since the concept of scale invariance has not been well recognized by policymakers but may help explain the persistent pattern of income distribution in Thailand. If income distribution in Thailand is found to be scale-invariant,

then it will not only cast doubt on the effectiveness of income redistributive policies and measures conducted during the past three decades but also post a challenge and make policymakers to think totally anew when designing and implementing policies and measures in order to change income distribution among different groups of population so that it will not become an obstacle to Thailand's economic and social development towards the SDGs.

## Materials and methods

To test the property of scale invariance in income distribution in Thailand, this study employs the actual data on income shares by quintile and by decile obtained from the NESDC [3] covering the period from 1988 to 2021. All data are provided in S1 Table. Note that the NESDC reports the data on income shares by quintile and by decile once every two years. Thus, there are total of 18 years of observations for each type of data groupings. In this study, scale invariance is defined as self-similarity in income distribution as represented by income shares by quintile and by decile across years. For notations, let $I_{y_i}^Q$ be quintile income share in year $i$ denoted as $y_i$ and $I_{y_i}^D$ be decile income share in year $i$ denoted as $y_i$, where $i$ = 1988, 1990, 1992, ..., 2021.

In order to test whether the distribution of income is scale-invariant, we have to demonstrate that the income distribution is statistically self-similar across years for each type of data groupings. Given that there is no agreement on the choice of statistical distributions that can be used to fit the actual income data and test the property of scale invariance as discussed in Introduction, this study uses an alternative method proposed by Sitthiyot et al. [16] which could be done by performing the pairwise K-S test to find out whether $I_{y_{1988}}^Q \cong I_{y_{1990}}^Q \cong I_{y_{1992}}^Q \cong$ ... $\cong I_{y_{2021}}^Q$ for income share by quintile and $I_{y_{1988}}^D \cong I_{y_{1990}}^D \cong I_{y_{1992}}^D \cong$ ... $\cong I_{y_{2021}}^D$ for income share by decile. As noted by Sitthiyot et al. [16], the K-S test is non-parametric and commonly used to determine if two datasets differ statistically. Its main advantages are that it makes no assumption about the distribution of data and works for small sample size. Provided that the number of observations ($n$) on income shares is 18, there are total of $\frac{n*(n-1)}{2} = 153$ pairs for each type of data groupings to be tested whether the respective income share in any given year is statistically different from the others. If the null hypothesis is not rejected, then it implies that the distribution of income is statistically scale-invariant or self-similar across years and vice versa. Tables 1 and 2 report the descriptive statistics of data on income shares by quintile and by decile of Thailand from 1988 and 2021.

## Results

The results from the pairwise K-S test based on the actual data on income share by quintile are reported in Table 3. They indicate that the distribution of income in Thailand from 1988 to 2021 is statistically scale-invariant or self-similar with *p-value* being equal to 1.000 in all 153 cases. This suggests that $I_{y_{1988}}^Q \cong I_{y_{1990}}^Q \cong I_{y_{1992}}^Q \cong$ ... $\cong I_{y_{2021}}^Q$ across years.

**Table 1. The descriptive statistics of data on income shares by quintile of Thailand from 1988 to 2021.**

| Quintile | Mean | Median | Mode | Minimum | Maximum | Standard deviation | No. of observations |
|----------|------|--------|------|---------|---------|--------------------|---------------------|
| Q1 | 0.045 | 0.043 | - | 0.038 | 0.055 | 0.005 | 18 |
| Q2 | 0.082 | 0.080 | - | 0.071 | 0.097 | 0.008 | 18 |
| Q3 | 0.126 | 0.124 | - | 0.111 | 0.143 | 0.010 | 18 |
| Q4 | 0.203 | 0.202 | - | 0.189 | 0.217 | 0.007 | 18 |
| Q5 | 0.545 | 0.548 | - | 0.489 | 0.590 | 0.029 | 18 |

**Table 2. The descriptive statistics of data on income shares by decile of Thailand from 1988 to 2021.**

| Decile | Mean | Median | Mode | Minimum | Maximum | Standard deviation | No. of observations |
|---|---|---|---|---|---|---|---|
| D1 | 0.016 | 0.016 | - | 0.011 | 0.021 | 0.002 | 18 |
| D2 | 0.028 | 0.027 | - | 0.024 | 0.035 | 0.004 | 18 |
| D3 | 0.037 | 0.035 | - | 0.031 | 0.044 | 0.004 | 18 |
| D4 | 0.046 | 0.045 | - | 0.039 | 0.053 | 0.004 | 18 |
| D5 | 0.056 | 0.055 | - | 0.049 | 0.064 | 0.005 | 18 |
| D6 | 0.070 | 0.069 | - | 0.062 | 0.079 | 0.005 | 18 |
| D7 | 0.088 | 0.087 | - | 0.080 | 0.096 | 0.005 | 18 |
| D8 | 0.115 | 0.115 | - | 0.109 | 0.120 | 0.003 | 18 |
| D9 | 0.162 | 0.163 | - | 0.151 | 0.171 | 0.005 | 18 |
| D10 | 0.382 | 0.391 | - | 0.331 | 0.424 | 0.026 | 18 |

**Table 3. The values of *D-statistic* from the pairwise K-S test based on the data on income shares by quintile with *p-values* shown in parentheses.**

| | 1988 | 1990 | 1992 | 1994 | 1996 | 1998 | 2000 | 2002 | 2004 | 2006 | 2007 | 2009 | 2011 | 2013 | 2015 | 2017 | 2019 | 2021 |
|---|---|---|---|---|---|---|---|---|---|---|---|---|---|---|---|---|---|---|
| **1988** | | 0.200 (1.000) | 0.200 (1.000) | 0.200 (1.000) | 0.200 (1.000) | 0.200 (1.000) | 0.200 (1.000) | 0.200 (1.000) | 0.200 (1.000) | 0.200 (1.000) | 0.200 (1.000) | 0.200 (1.000) | 0.200 (1.000) | 0.200 (1.000) | 0.200 (1.000) | 0.200 (1.000) | 0.200 (1.000) | 0.200 (1.000) |
| **1990** | | | 0.200 (1.000) | 0.200 (1.000) | 0.200 (1.000) | 0.200 (1.000) | 0.200 (1.000) | 0.200 (1.000) | 0.200 (1.000) | 0.200 (1.000) | 0.200 (1.000) | 0.200 (1.000) | 0.200 (1.000) | 0.200 (1.000) | 0.200 (1.000) | 0.200 (1.000) | 0.200 (1.000) | 0.200 (1.000) |
| **1992** | | | | 0.200 (1.000) | 0.200 (1.000) | 0.200 (1.000) | 0.200 (1.000) | 0.200 (1.000) | 0.200 (1.000) | 0.200 (1.000) | 0.200 (1.000) | 0.200 (1.000) | 0.200 (1.000) | 0.200 (1.000) | 0.200 (1.000) | 0.200 (1.000) | 0.200 (1.000) | 0.200 (1.000) |
| **1994** | | | | | 0.200 (1.000) | 0.200 (1.000) | 0.200 (1.000) | 0.200 (1.000) | 0.200 (1.000) | 0.200 (1.000) | 0.200 (1.000) | 0.200 (1.000) | 0.200 (1.000) | 0.200 (1.000) | 0.200 (1.000) | 0.200 (1.000) | 0.200 (1.000) | 0.200 (1.000) |
| **1996** | | | | | | 0.200 (1.000) | 0.200 (1.000) | 0.200 (1.000) | 0.200 (1.000) | 0.200 (1.000) | 0.200 (1.000) | 0.200 (1.000) | 0.200 (1.000) | 0.200 (1.000) | 0.200 (1.000) | 0.200 (1.000) | 0.200 (1.000) | 0.200 (1.000) |
| **1998** | | | | | | | 0.200 (1.000) | 0.200 (1.000) | 0.200 (1.000) | 0.200 (1.000) | 0.200 (1.000) | 0.200 (1.000) | 0.200 (1.000) | 0.200 (1.000) | 0.200 (1.000) | 0.200 (1.000) | 0.200 (1.000) | 0.200 (1.000) |
| **2000** | | | | | | | | 0.200 (1.000) | 0.200 (1.000) | 0.200 (1.000) | 0.200 (1.000) | 0.200 (1.000) | 0.200 (1.000) | 0.200 (1.000) | 0.200 (1.000) | 0.200 (1.000) | 0.200 (1.000) | 0.200 (1.000) |
| **2002** | | | | | | | | | 0.200 (1.000) | 0.200 (1.000) | 0.200 (1.000) | 0.200 (1.000) | 0.200 (1.000) | 0.200 (1.000) | 0.200 (1.000) | 0.200 (1.000) | 0.200 (1.000) | 0.200 (1.000) |
| **2004** | | | | | | | | | | 0.200 (1.000) | 0.200 (1.000) | 0.200 (1.000) | 0.200 (1.000) | 0.200 (1.000) | 0.200 (1.000) | 0.200 (1.000) | 0.200 (1.000) | 0.200 (1.000) |
| **2006** | | | | | | | | | | | 0.200 (1.000) | 0.200 (1.000) | 0.200 (1.000) | 0.200 (1.000) | 0.200 (1.000) | 0.200 (1.000) | 0.200 (1.000) | 0.200 (1.000) |
| **2007** | | | | | | | | | | | | 0.200 (1.000) | 0.200 (1.000) | 0.200 (1.000) | 0.200 (1.000) | 0.200 (1.000) | 0.200 (1.000) | 0.200 (1.000) |
| **2009** | | | | | | | | | | | | | 0.200 (1.000) | 0.200 (1.000) | 0.200 (1.000) | 0.200 (1.000) | 0.200 (1.000) | 0.200 (1.000) |
| **2011** | | | | | | | | | | | | | | 0.200 (1.000) | 0.200 (1.000) | 0.200 (1.000) | 0.200 (1.000) | 0.200 (1.000) |
| **2013** | | | | | | | | | | | | | | | 0.200 (1.000) | 0.200 (1.000) | 0.200 (1.000) | 0.200 (1.000) |
| **2015** | | | | | | | | | | | | | | | | 0.200 (1.000) | 0.200 (1.000) | 0.200 (1.000) |
| **2017** | | | | | | | | | | | | | | | | | 0.200 (1.000) | 0.200 (1.000) |
| **2019** | | | | | | | | | | | | | | | | | | 0.200 (1.000) |
| **2021** | | | | | | | | | | | | | | | | | | |

**Table 4. The values of *D-statistic* from the pairwise K-S test based on the data on income shares by decile with *p-values* shown in parentheses.**

| | 1988 | 1990 | 1992 | 1994 | 1996 | 1998 | 2000 | 2002 | 2004 | 2006 | 2007 | 2009 | 2011 | 2013 | 2015 | 2017 | 2019 | 2021 |
|---|---|---|---|---|---|---|---|---|---|---|---|---|---|---|---|---|---|---|
| **1988** | | 0.100 (1.000) | 0.100 (1.000) | 0.100 (1.000) | 0.100 (1.000) | 0.100 (1.000) | 0.100 (1.000) | 0.100 (1.000) | 0.100 (1.000) | 0.100 (1.000) | 0.100 (1.000) | 0.100 (1.000) | 0.100 (1.000) | 0.100 (1.000) | 0.100 (1.000) | 0.100 (1.000) | 0.100 (1.000) | 0.100 (1.000) |
| **1990** | | | 0.100 (1.000) | 0.100 (1.000) | 0.100 (1.000) | 0.100 (1.000) | 0.100 (1.000) | 0.100 (1.000) | 0.100 (1.000) | 0.100 (1.000) | 0.100 (1.000) | 0.100 (1.000) | 0.100 (1.000) | 0.100 (1.000) | 0.200 (0.988) | 0.100 (1.000) | 0.200 (0.988) | 0.200 (0.988) |
| **1992** | | | | 0.100 (1.000) | 0.100 (1.000) | 0.100 (1.000) | 0.100 (1.000) | 0.100 (1.000) | 0.100 (1.000) | 0.100 (1.000) | 0.100 (1.000) | 0.100 (1.000) | 0.100 (1.000) | 0.200 (0.988) | 0.200 (0.988) | 0.200 (0.988) | 0.200 (0.988) | 0.200 (0.988) |
| **1994** | | | | | 0.100 (1.000) | 0.100 (1.000) | 0.100 (1.000) | 0.100 (1.000) | 0.100 (1.000) | 0.100 (1.000) | 0.100 (1.000) | 0.100 (1.000) | 0.100 (1.000) | 0.100 (1.000) | 0.200 (0.988) | 0.100 (1.000) | 0.200 (0.988) | 0.200 (0.988) |
| **1996** | | | | | | 0.100 (1.000) | 0.100 (1.000) | 0.100 (1.000) | 0.100 (1.000) | 0.100 (1.000) | 0.100 (1.000) | 0.100 (1.000) | 0.100 (1.000) | 0.100 (1.000) | 0.200 (0.988) | 0.100 (1.000) | 0.200 (0.988) | 0.200 (0.988) |
| **1998** | | | | | | | 0.100 (1.000) | 0.100 (1.000) | 0.100 (1.000) | 0.100 (1.000) | 0.100 (1.000) | 0.100 (1.000) | 0.100 (1.000) | 0.100 (1.000) | 0.100 (1.000) | 0.100 (1.000) | 0.200 (0.988) | 0.200 (0.988) |
| **2000** | | | | | | | | 0.100 (1.000) | 0.100 (1.000) | 0.100 (1.000) | 0.100 (1.000) | 0.100 (1.000) | 0.100 (1.000) | 0.100 (1.000) | 0.200 (0.988) | 0.200 (0.988) | 0.200 (0.988) | 0.200 (0.988) |
| **2002** | | | | | | | | | 0.100 (1.000) | 0.100 (1.000) | 0.100 (1.000) | 0.100 (1.000) | 0.100 (1.000) | 0.100 (1.000) | 0.100 (1.000) | 0.100 (1.000) | 0.200 (0.988) | 0.200 (0.988) |
| **2004** | | | | | | | | | | 0.100 (1.000) | 0.100 (1.000) | 0.100 (1.000) | 0.100 (1.000) | 0.100 (1.000) | 0.100 (1.000) | 0.100 (1.000) | 0.100 (1.000) | 0.100 (1.000) |
| **2006** | | | | | | | | | | | 0.100 (1.000) | 0.100 (1.000) | 0.100 (1.000) | 0.100 (1.000) | 0.100 (1.000) | 0.100 (1.000) | 0.200 (0.988) | 0.200 (0.988) |
| **2007** | | | | | | | | | | | | 0.100 (1.000) | 0.100 (1.000) | 0.100 (1.000) | 0.100 (1.000) | 0.100 (1.000) | 0.100 (1.000) | 0.100 (1.000) |
| **2009** | | | | | | | | | | | | | 0.100 (1.000) | 0.100 (1.000) | 0.100 (1.000) | 0.100 (1.000) | 0.100 (1.000) | 0.100 (1.000) |
| **2011** | | | | | | | | | | | | | | 0.100 (1.000) | 0.100 (1.000) | 0.100 (1.000) | 0.100 (1.000) | 0.100 (1.000) |
| **2013** | | | | | | | | | | | | | | | 0.100 (1.000) | 0.100 (1.000) | 0.100 (1.000) | 0.100 (1.000) |
| **2015** | | | | | | | | | | | | | | | | 0.100 (1.000) | 0.100 (1.000) | 0.100 (1.000) |
| **2017** | | | | | | | | | | | | | | | | | 0.100 (1.000) | 0.100 (1.000) |
| **2019** | | | | | | | | | | | | | | | | | | 0.100 (1.000) |
| **2021** | | | | | | | | | | | | | | | | | | |

For the actual data on income share by decile, the pairwise K-S test results as reported in Table 4 show almost identical findings in that the income distribution in Thailand from 1988 to 2021 has a property of scale invariance or self-similarity. In all 153 cases, the *p-value* ranges between 0.988 and 1.000, indicating that there is no statistically difference in income distribution across years. That is $I^{D}_{y_{1988}} \cong I^{D}_{y_{1990}} \cong I^{D}_{y_{1992}} \cong \ldots \cong I^{D}_{y_{2021}}$.

These empirical findings confirm that the patterns of income distribution, represented by the income shares by quintile and by decile, had not significantly changed during the past three decades as illustrated earlier in Figs 1 and 2. They also confirm the empirical evidence on scale invariance or self-similarity in income and wealth distributions found in previous studies [8–12, 14–16] as discussed in Introduction. However, with an exception of the study by Sitthiyot et al. [16], these studies [8–12, 14, 15] test the property of scale invariance or self-similarity by fitting different complicated statistical distributions to the actual income observations. This study uses the pairwise K-S test introduced by Sitthiyot et al. [16] which is relatively simple and does not require a priori specification with regard to the distribution of income data but comes up with similar findings.

## Discussion

The development of Thailand has advanced the country in many aspects, creating occupational and income securities that have led to considerable poverty reduction [4]. Despite such achievements, the progress in resolving the issue of income distribution that had been skewed towards a small percentage of the population is somewhat slow and considered a challenging issue [4]. While the concept of scale invariance has not been well recognized by policymakers and countless efforts have been continuously put in trying to change the pattern of income distribution in Thailand, the empirical results based on the pairwise K-S test from this study show that the pattern of income distribution in Thailand from 1988 to 2021 remains statistically unchanged with *p-values* ranging between 0.988 and 1.000. These findings imply that income redistributive policies and measures that had been implemented in the past three decades are not quite up to the task as illustrated by the income shares by quintile and by decile that do not statistically differ across years.

Given that scale invariance is not a common expression and the term like self-similarity is hardly used in economics [7], the empirical findings from this study indicate that it is important for policymakers to have a clear understanding and acknowledge the property of scale invariance or self-similarity in income distribution. In addition, policymakers have to understand that when it comes to policy design and implementation with an aim to change the distribution of income that had remained almost unchanged for several decades, it requires creativity, openness to new ways of thinking, and a large amount of disciplined analyses [31]. Even though the property of scale invariance suggests that the underlying generic features and physical principles of income distribution are independent of detailed dynamics and/or specific characteristics of the economic system, from our viewpoint, it does not mean that the income distribution cannot be changed. This study would like to propose that the change in income distribution might have to be like the phase transition of substance in physics.

To give a simple analogy, we could think of income distribution as the state of water being liquid, the detailed dynamics and/or specific characteristics of the economic system as the environs, and whatever we do in order to change the environs as policies and measures. Examples of the detailed dynamics and/or specific characteristics of the economic system mainly include form of governance, institutional arrangement, administration of justice, deployment of technology, and medium of exchange. As long as the temperature is between 0 and 100 degrees Celsius, whatever we do to the environs under normal pressure will not change the generic features and physical principles of water. In order to change water from the liquid state to the gaseous state, we have to supply enough energy to the environs so that the temperature reaches 100 degrees Celsius.

The same logic could be applied to income distribution. In order to change the generic features and physical principles of income distribution, it requires policies and measures that can change the detailed dynamics and/or specific characteristics of the economic system as specified above which in turn cause the phase transition of income distribution. This idea might seem far-fetched but is worth considering given that the conventional development policies and measures conducted during the past three decades had not been quite effective in changing the pattern of income distribution in Thailand. Moreover, the income distribution which is skewed towards a small percentage of population still remains a challenging issue for the development of Thailand towards the SDGs. How to change form of governance, institutional arrangement, administration of justice, deployment of technology, and medium of exchange that could bring about the phase transition of income distribution remains uncharted territory.

## Supporting information

**S1 Table. Data on income shares by quintile and by decile of Thailand from 1988 to 2021.** (PDF)

## Acknowledgments

TS is grateful to Dr. Suradit Holasut for guidance and comments.

## Author Contributions

**Conceptualization:** Thitithep Sitthiyot.

**Formal analysis:** Thitithep Sitthiyot.

**Methodology:** Thitithep Sitthiyot.

**Validation:** Thitithep Sitthiyot.

**Writing – original draft:** Thitithep Sitthiyot.

**Writing – review & editing:** Thitithep Sitthiyot.

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
