## [Decision Letter · Decision Letter 0]

7 Jun 2023

PONE-D-23-12286Income distribution in Thailand is scale-invariantPLOS ONE

Dear Dr. Sitthiyot,

Thank you for submitting your manuscript to PLOS ONE. After careful consideration, we feel that it has merit but does not fully meet PLOS ONE’s publication criteria as it currently stands. Therefore, we invite you to submit a revised version of the manuscript that addresses the points raised during the review process.

We look forward to receiving your revised manuscript.

Kind regards,

Ricky Chee Jiun Chia

Academic Editor

PLOS ONE

Journal Requirements:

Reviewers' comments:

Reviewer's Responses to Questions

**Comments to the Author**

1. Is the manuscript technically sound, and do the data support the conclusions?

Reviewer #1: Yes

2. Has the statistical analysis been performed appropriately and rigorously? 

Reviewer #1: Yes

3. Have the authors made all data underlying the findings in their manuscript fully available?

Reviewer #1: Yes

4. Is the manuscript presented in an intelligible fashion and written in standard English?

Reviewer #1: Yes

5. Review Comments to the Author

Reviewer #1: Using the actual data on income shares by quintile and by decile of Thailand from 1988 to 2021, this study examines

whether or not the income distribution has a property of scale invariance across years.

Comments

Overall, I find the study interesting and worthy of investigation. However, I have several comments if addressed will improve the paper further and I outline them below.

0) Abstract. The abstract should be shortened. For publishing the paper, I suggest that author(s) deliver a concise and factual abstract. The abstract should state briefly the purpose of the research, the principal results and major conclusions. Readers are usually interested in highlighting ideas or arguments that are attractively presented in the abstract. For example, delete “…The concept of scale invariance or self-similarity is not well recognized by policymakers but may help explain the pattern of income distribution in Thailand that had not markedly changed over the past three decades...This requires creativity, openness to new ways of thinking, and disciplined analyses when it comes to policy design and implementation…”.

1) Although the authors make good attempt in providing a compelling case for the need of this study, but it can be strengthened further by spelling out more succinctly why the need to explore the nexus between the property of scale invariance and income distribution. This must come out clearly, as its present form, it has not been clearly articulated on.

2) The contribution section appears sketchy and general. I would like to see the contribution section strengthened drawing from the findings of the study.

3) Related literature review with Thailand data is not clear. The author(s) should cite relevant related papers. For example, the study

https://doi.org/10.1016/j.qref.2004.02.003

employ Thailand (and Asian) economic data.

4) It is not properly motivated the use of the pairwise K-S test.

6. PLOS authors have the option to publish the peer review history of their article (what does this mean?). If published, this will include your full peer review and any attached files.

Reviewer #1: No

---

## [Author Response · Author response to Decision Letter 0]

21 Jun 2023

Manuscript Number: PONE-D-23-12286R1

Title: Income distribution in Thailand is scale-invariant

I sincerely thank Reviewer #1 for providing several important comments and useful suggestions. I would like to inform Reviewer #1 that I have made a revision according to comments and suggestions made by Reviewer #1. I hope that my revised manuscript is clearer in all aspects that Reviewer #1 has commented and/or suggested. Let me respond to Reviewer #1’s comments and suggestions as follows.

- Overall, I find the study interesting and worthy of investigation. However, I have several comments if addressed will improve the paper further and I outline them below.

I would like to thank Reviewer #1 for finding my paper interesting and worthy of investigation. I really appreciate it.

- Abstract. The abstract should be shortened. For publishing the paper, I suggest that author(s) deliver a concise and factual abstract. The abstract should state briefly the purpose of the research, the principal results and major conclusions. Readers are usually interested in highlighting ideas or arguments that are attractively presented in the abstract. For example, delete “…The concept of scale invariance or self-similarity is not well recognized by policymakers but may help explain the pattern of income distribution in Thailand that had not markedly changed over the past three decades...This requires creativity, openness to new ways of thinking, and disciplined analyses when it comes to policy design and implementation…”.

I follow Reviewer #1’s advice by deleting unnecessary sentences as suggested by Reviewer #1 above and only stating the purpose of the study, the principal results, major conclusions, and policy implications in Abstract in my revised manuscript.

- Although the authors make good attempt in providing a compelling case for the need of this study, but it can be strengthened further by spelling out more succinctly why the need to explore the nexus between the property of scale invariance and income distribution. This must come out clearly, as its present form, it has not been clearly articulated on.

In response to Reviewer #1’s useful comment on this point, I explain the need to explore the link between scale invariance and income distribution in Introduction, paragraph 5 in my revised manuscript.

- The contribution section appears sketchy and general. I would like to see the contribution section strengthened drawing from the findings of the study.

I would like to thank Reviewer #1 for this very useful comment. I rewrote Discussion by first summarizing the findings and then trying to draw policy implications from the findings as stated in Discussion, paragraphs 1 and 2 in my revised manuscript. I also add suggestion for future research in Discussion, paragraph 4 in my revised manuscript. 

- Related literature review with Thailand data is not clear. The author(s) should cite relevant related papers. For example, the study https://doi.org/10.1016/j.qref.2004.02.003 employ Thailand (and Asian) economic data.

In response to Reviewer #1 on this issue, I would like to clarify with Reviewer #1 that I obtained the data on income shares from the Office of National Economic and Social Development Council (NESDC) website which is cited in References in my revised manuscript. To make the data used in this study accessible and transparent, I also provide all data on income shares by quintile and by decile of Thailand from 1988 to 2021 in Supporting Information, S1 Table.

I would like to note that, although the NESDC website has an English language version, the income data can be accessed from its website in Thai language version only. The link to the NESDC website, which is https://www.nesdc.go.th/main.php?filename=PageSocial, is included in References in my revised manuscript. If readers go to this link and click “สถิติด้านความยากจนและการกระจายรายได้” which can be translated as “Poverty and income distribution statistics”, readers can download the data on income shares by quintile and by decile of Thailand from 1988 to 2021 which are in Tables 8.2 and 8.5. Please see the attached file (Response to Reviewers_R1) for the step-by-step of how to access the data.

- It is not properly motivated the use of the pairwise K-S test.

In response to Reviewer #1’s comment on the use of pairwise K-S test, in Introduction, paragraph 4 in my revised manuscript, besides the study by Chakrabarti et al. (2013) who note that there tends to be an agreement among academic scholars in economics, statistics, and physics that the upper tail of income distribution could be well described by power law but the lower part of income distribution could be fitted by power law, gamma, or log-normal distribution, I include additional studies, namely, Montroll and Shlesinger (1982), Champernowne and Cowell (1998), Drăgulescu and Yakovenko (2001), Chatterjee and Chakrabarti (2007), and Yakovenko and Barkley Rosser (2009), all of which use complicated statistical distributions in order to fit the income data and test the property of scale invariance as a ground for using the pairwise K-S test which is relatively simpler and does not require a priori assumption with regard to the distribution of data but could achieve the same task.

In addition, in Materials and Methods, paragraph 2, I explain the reason for using the K-S test by referring to the study by Sitthiyot et al. (2020) who note that the K-S test is non-parametric and commonly used to determine if two datasets differ statistically. Its main advantages are that it makes no assumption about the distribution of data and works for small sample size.

---

## [Editor Report · Decision Letter 1]

22 Jun 2023

Income distribution in Thailand is scale-invariant

PONE-D-23-12286R1

Dear Dr. Thitithrp Sitthiyot,

We’re pleased to inform you that your manuscript has been judged scientifically suitable for publication and will be formally accepted for publication once it meets all outstanding technical requirements.

Kind regards,

Ricky Chee Jiun Chia

Academic Editor

PLOS ONE
---

## [Editor Report · Acceptance letter]

3 Jul 2023

PONE-D-23-12286R1 

Income distribution in Thailand is scale-invariant 

Dear Dr. Sitthiyot:

I'm pleased to inform you that your manuscript has been deemed suitable for publication in PLOS ONE. Congratulations! Your manuscript is now with our production department. 

Kind regards, 

on behalf of

Dr. Ricky Chee Jiun Chia 

Academic Editor

PLOS ONE